# [Re] Identifying Through Flows for Recovering Latent Representations

**Max van den Heuvel**
Institute of Informatics
University of Amsterdam
max.heuvel@live.nl

**Roel Klein**
Institute of Informatics
University of Amsterdam
roelklein99@gmail.com

**Tim Stolp**
Institute of Informatics
University of Amsterdam
tim2stolp@gmail.com

**Fengyuan Sun**
Institute of Informatics
University of Amsterdam
fengyuan.sun@student.uva.nl

## Reproducibility Summary

**Scope of Reproducibility**

iFlow is more identifiable than iVAE is the main claim of the original paper. Other claims proposed are the preservation of geometry of latent sources, more accurate data modelling, and consistent identifiability across dimensions.

**Methodology**

The author's code was available and used to generate synthetic data and train the iFlow models, requiring only minor adjustments. Reproduced results were compared to the papers results and the model was extended upon to further test the robustness of their claims. The models were trained on an Nvidia RTX 2080 and using around 50+ GPU hours.

**Results**

The reproduced MCC scores in 2D do not support the claim that iFlow yields identifiability improvements over iVAE due to iVAE scoring higher. iFlow might preserve the geometry of the latent space. In contradiction to the paper, there was no case where iVAE collapsed. The results of our iFlow performance study are almost identical to that of the original paper suggesting that iFlow indeed yields improved identifiability and modelling of data distributions. The results of the final reproduction suggests in favour of the improved identifiability across dimensions of latent space. Using the best performing seed, iFlow scored around 0.25 higher on the correlation coefficient. Our extended experiments show more stability of iFlow when increasing the data complexity.

**What was easy**

Code was provided, making training the models generally straightforward. The code was well-structured and included descriptions of the hyperparameters. The paper provided sufficient information on the hyperparameters used.

**What was difficult**

No code was provided for creating the figures of the paper. Creating these ourselves required a thorough understanding of both the paper and the code. One experiment also required uncommenting some lines of code, which was changed to make the code more dynamic. Implementing a generative model using the iFlow framework proved difficult, because there is currently no method to generate new samples.

**Communication with original authors**

We were unable to come in contact with the authors.

33rd Conference on Neural Information Processing Systems (NeurIPS 2020), Vancouver, Canada.

# 1 Introduction

Representation learning focuses on learning representations that can capture the underlying probability distribution of the explanatory features of the data. A subgoal is identifiability, which refers to the recovery of the true latent variables of the underlying data distribution. Representation learning is widely adopted in deep generative models such as variational autoencoders (VAE) (Kingma and Welling, 2014), generative adverserial networks (Goodfellow et al., 2014) and flow-based generative models. However, these approaches do not take the property of identifiability into account. In a more recent study, Khemakhem et al., 2020 introduced an identifiability theory for generative models and applied it to the VAE, proposing the identifiable iVAE. One downside of this framework is that VAE optimizes a lower bound of the log-likelihood, which potentially causes the model to be less identifiable. Following this theory, Li, Hooi, and Lee, 2020 presented a novel identifiable framework using a flow-based model for estimating latent representations, which is able to maximize the likelihood of the data directly, resulting in higher identifiability over iVAE.

In this work, the main results from the "Identifying Through Flows for Recovering Latent Representations" paper are reproduced (Li, Hooi, and Lee, 2020). The code for the original implementation is available on GitHub[1]. The experiments are recreated based on the details listed in the paper. As an extension, the robustness of iFlow compared to iVAE is tested on datasets with higher complexity. Furthermore, an attempt was made to apply iFlow on the MNIST dataset (LeCun and Cortes, 2010).

# 2 Background

In statistical machine learning it is often necessary to explain the process that generates the observations. This requires an estimation of the often complex densities. Normalizing Flows are a tool to fit these complex densities by applying a series of transformation on a base distribution. A normalizing flow is a diffeomorphism between two equivalent spaces (i.e. an invertible and differentiable transformation with a differentiable inverse). These flows deal with the marginal likelihood with exact inference and are a transform of a tractable distribution into a complex distribution using invertible and differential mappings. The challenge lies in designing a normalizing flow which has an inverse that is efficient to calculate without sacrificing its capabilities of mapping simple distributions to complex ones (Li, Hooi, and Lee, 2020).

Under these conditions the density of an observed random variable $\mathbf{x}$ can be obtained using the change of variable formula:

$$p_X(\mathbf{x}) = p_Z(\mathbf{h}(\mathbf{x})) \left| \det\left(\frac{\partial \mathbf{h}}{\partial \mathbf{x}}\right) \right| = p_Z(\mathbf{z}) \left| \det\left(\frac{\partial \mathbf{f}}{\partial \mathbf{z}}\right) \right|^{-1}, \tag{1}$$

where $\mathbf{h}$ is the inverse of the normalizing flow $\mathbf{f}$. To map a complex nonlinear invertible bijection, a series of such functions can be composed. The optimization of the diffeomorphisms parameters is then defined by the maximization of the log-likelihood for the density estimation model:

$$\boldsymbol{\phi}^* = \arg\max_{\phi} \mathbb{E}_{\mathbf{x}} \left[ \log p_Z(\mathbf{h}(\mathbf{x}; \boldsymbol{\phi})) + \log \left| \det\left(\frac{\partial \mathbf{h}(\mathbf{x}; \boldsymbol{\phi})}{\partial \mathbf{x}}\right) \right| \right]. \tag{2}$$

The theoretical guarantee of invertibility and expressiveness of normalizing flows can be used to recover the true mixing mapping from sources to observations and achieving identifiability. By aligning normalizing flows with this identifiability theory it is desirable to learn a latent-variable model with identifiability guarantees as done by Li, Hooi, and Lee, 2020.

Identifiability is thus achieved using a conditionally factorized prior distribution over the latent variables using an auxiliary variable, which can be an additionally observed variable among other things (Khemakhem et al., 2020). The conditional generative model is assumed to be a factorized exponential family conditioned on the auxiliary variable $\mathbf{u}$. The probability density function follows:

$$p_{\mathbf{T}, \boldsymbol{\lambda}}(\mathbf{z}|\mathbf{u}) = \prod_{i=1}^{n} p_i(\mathbf{z}_i|\mathbf{u}) = \prod_i \frac{Q_i(\mathbf{z}_i)}{Z_i(\mathbf{u})} \exp\left[\sum_{j=1}^{k} T_{i,j}(z_i)\lambda_{i,j}(\mathbf{u})\right], \tag{3}$$

where $Q_i$ is the base measure set to one for simplicity, $Z_i(\mathbf{u})$ is the normalizing constant, $T_{i,j}$ are the components of sufficient statistics, which is set to $z^2$ and $z$ with $k$ indicating the maximum order of statistics set to two. Finally $\lambda_{i,j}(\mathbf{u})$ are the natural parameters depending on $\mathbf{u}$ parameterized by a multi-layer perceptron with learnable parameters. The identifiability of the latent-variable family is then given by *Theorem 4.1* in Li, Hooi, and Lee, 2020.

---

[1]Code for the original implementation can be found here `https://github.com/MathsXDC/iFlow`

This leads to the optimization objective, minimizing the negative log-likelihood:

$$\mathcal{L}(\Theta) = \mathbb{E}_{\mathbf{x},\mathbf{u} \sim p_D} \left[ \left( \sum_{i=1}^{n} \log Z_i(\mathbf{u}) \right) - \text{trace}\left( \mathbf{T}(\mathbf{z})\boldsymbol{\lambda}(\mathbf{u})^T \right) - \log \left| \det\left( \frac{\partial \mathbf{h}_\phi}{\partial \mathbf{x}} \right) \right| \right], \tag{4}$$

where $p_D$ denotes the empirical distribution. The first two components of this objective are derived from the exponential family distribution (formula 3). The third component is derived from the change of volume of the normalizing flow, where $\mathbf{h}_\phi$ is a normalizing flow of any kind (Li, Hooi, and Lee, 2020).

## 3   Scope of reproducibility

iFlow is a framework for creating identifiable normalizing flows, applying the ideas behind iVAE to normalizing flows. In order to make claims about the identifiability, both models were tested on a synthetic dataset with known latent sources, specifically a non-stationary Gaussian time-series for latent sources, from which observations where generated by putting them through an MLP, as described in Khemakhem et al., 2020. The authors have made the following claims regarding the performance of iFlow compared to iVAE on this synthetic dataset:

- iFlow yields improved identifiability, preserving the geometry of the latent sources.
- iFlow gives a significantly higher likelihood over the data, showing that iFlow models the data distribution more accurately.
- iFlow's improvement in identifiability is consistent across dimensions of the latent space.

This work is planned to be extended by performing an ablation study of iFlow and iVAE on increased dataset complexity. This leads to the claim:

- iFlow has more consistent performance on data with increased complexity than iVAE.

To test this claim, iFlow is applied on a real dataset, specifically MNIST, rather than synthetic data. However, this proved to be more difficult than expected, which is described in section 6.2. As a result, no claims can be made yet on the performance of iFlow on a real dataset.

## 4   Methodology

To reproduce the paper's results, the corresponding code, available on the paper's GitHub page, was used to generate the data sets and train the iFlow models. The descriptions of the arguments of the code were clear enough to be able to understand and run it. Furthermore the code base contained the scripts that were used to reproduce the paper's results. The computer systems that were used consisted of both the Linux and Windows operating systems and the Nvidia GTX 950M and RTX 2080 GPUs respectively. For the larger models, the GTX 950M did not have enough memory. The code initially was not compatible with Windows due to the absence of batch scripts and usage of a linux specific file locking library. The scripts were written and the file locking library was replaced with a cross-platform one called Portalocker. Finally some file names were changed to not include illegal characters according to Windows file naming conventions.

### 4.1   Model descriptions

The iFlow model consists of a multi-layer perceptron to model the natural parameters of the conditional factorized exponential distribution where the last layer is followed by a softplus non-linearity. This negative activation function is used on the second order parameters to ensure their finiteness. The bijections of the flows are modeled by RQ-NSF(AR) (Durkan et al., 2019) with flow length of 10 and bin size of 8. In total, iFlow contains 28350 parameters. This ensures enough flexibility and expressiveness (Li, Hooi, and Lee, 2020). The Adam optimizer was used to train the model in combination with the ReduceLROnPlateau learning rate scheduler. The implementation of the iVAE model is adapted from the codebase[2] of its respective authors, Khemakhem et al., 2020.

### 4.2   Datasets

We have create synthetic data using the same configurations as in the original paper (Li, Hooi, and Lee, 2020), following the provided code. The datasets used to replicate the main experiments consist of uncentered data of 40 segments,

---

[2]https://github.com/siamakz/iVAE

1000 observations per segment, a latent source dimension of 5, a data dimension of 5, 3 layers in the mixing MLP, 60 different seeds, a Gaussian prior distribution for the sources, and xtanh activation functions for the mixing MLP. This results in data sets containing observations and their latent sources. Furthermore, these sources are mixed according to Hyvarinen, Sasaki, and Turner, 2019 using the MLP. The auxiliary variable is set as the one hot encoded index of the segment. For visualizing the 2D cases, the synthesized dataset consists of 5 segments, a latent source dimensionality of 2, data dimensionality of 2 and otherwise equal parameters.

For an extension of the model the MNIST[3] handwritten digit database was used (LeCun and Cortes, 2010), which is a well-known standard for image classification and generation.

### 4.3 Hyperparameters

The same hyperparameters were used as provided in the scripts present in the GitHub code base. These consisted of a learning rate of 1e-3, a learning rate drop factor of 0.25, and a learning rate scheduler patience of 10. The training was done for 20 epochs with a batch size of 64, as was mentioned in the original report. However, due to a bug in the provided code the 2D visualizations could initially only use a batch size of 8. This problem was resolved and the final results also used a batch size of 64 for all experiments.

### 4.4 Experimental setup

We reproduced the three main experiments of the paper following their described methodology and description of results. Not all experimentation details were included in the paper which lead to implementation assumptions which will be outlined in the following subsections. The code for all reproduced experiments is available on GitHub [4].

#### 4.4.1 Metrics

To get a numerical evaluation of identifiability, model performance on the synthetic dataset is measured by calculating the mean correlation coefficient (MCC) between the predicted latents and the corresponding original latents. This is carried out by first computing the correlation coefficients between all predicted and source latent dimensions. Then, to revert the permutations in latent space, a linear sum assignment problem is solved and each predicted latent component is assigned to its most correlated source component. A high MCC score indicates high identifiability of the model in recovering the true latent sources.

#### 4.4.2 Identifiability in two dimensional space

The first experiment aims to both test and visualize the identifiability of iFlow for two dimensional cases, so for the case where the dimensionality of the latent sources and the observations is two. The original latent sources and observations are saved, as well as the predictions of iVAE and iFlow models. These are plotted alongside each other, visualizing the identifiability of both models. Additionally, the MCC between the model predictions and the latent sources is calculated, which gives a numerical evaluation of the identifiability. These visualizations are done for ten different data generation seeds.

#### 4.4.3 iFlow performance

For the second experiment, a script is provided by the authors that runs the program for 100 seeds. For this paper only 60 seeds were used due to computational time constraints. A dataset is generated per seed followed by the training of an iFlow model on this dataset. To evaluate the model, the MCC is calculated between the original sources of the data and the estimations of the model. A high correlation would indicate good estimation of the latent sources. Additionally, the absolute model loss for each seed configuration is measured.

#### 4.4.4 Correlation across latent dimensions

The third experiment aimed to quantify the models performance across dimensions of the latent space. The original paper did not extensively test this claim, but only compared correlation scores of one seed for iFlow and iVAE. To reproduce this, iFlow and iVAE models are trained on seed 49, which empirically proved to be the best performing seed and most likely the seed which is reported by the authors. Afterwards, the correlation coefficients for each of the 5 dimensions are calculated between the estimation of the latents and source latents of each datapoint. For visualization

---

[3]MNIST dataset can be downloaded here `http://yann.lecun.com/exdb/mnist/`

[4]The code created for the experiments in this paper can be found here `https://github.com/TimStolp/iFlow`.

purposes, the mean estimated and source signals are visualized. These signals are standardized to be better visually comparable.

To further study the validity of the third claim, we compared the means and standard deviations of the correlation coefficients of iFlow and iVAE for each dimension, averaged over 60 seeds.

### 4.4.5 Performance with increased data complexity

To extend the papers research, an investigation on the performance of iFlow and iVAE on changing data complexity was performed. This was done by increasing the number of layers in the mixing MLP. The MCC performances were measured across ten seeds on three different dataset configurations. For these configurations the number of MLP layers were three, four and five. The used dataset consisted of five different segments and 1000 points each, with latent and observation dimensions set to two.

### 4.4.6 Computational requirements

With the setup described above, training a single iFlow model with an RTX 2080 GPU takes around 30 minutes, whereas training iVAE takes approximately 2 minutes. The data generation and the calculation of the MCCs are fast and can be done in a matter of seconds. The total GPU hours spent is around 50+ hours. Training without a GPU is extremely slow and not recommended.

### 4.4.7 Extension to MNIST attempt

Training the model on MNIST required some extensions to the given code such as changing the dataloaders, removing the requirement for source latent values and handling auxiliary variables over 10 categories. When modeling discrete observations, the mapping from the latent variable to the source can no longer be injective which means the key assumption of the identifiability theory no longer holds (Khemakhem et al., 2020). Since MNIST consists of discrete data, dequantization is necessary, for which uniform noise was used followed by an inverse sigmoid.

In order to sample from the generated distribution, we tried inverse transform sampling. This is a method for generating samples at random from a probability distribution given the inverse cumulative distribution function (CDF). To achieve the CDF, the indefinite integral of the exponential family distribution (Formula 3) was taken:

$$\int p_{\mathbf{T},\boldsymbol{\lambda}}(\mathbf{z}|\mathbf{u})dz = \frac{1}{Z_i(\mathbf{u})}\frac{1}{2\sqrt{\xi}}\sqrt{\pi}e^{-\eta^2/(4\xi)}\mathrm{erfi}(\frac{2\xi z + \eta}{2\sqrt{\xi}}), \tag{5}$$

where $Z_i(\mathbf{u})$ is the normalization constant, $z$ is the desired source, $\xi$ and $\tau$ are the parameters learned by an MLP, with $\tau$ being strictly negative, and erfi is the imaginary error function. To then sample, the inverse of this CDF is calculated:

$$z = \frac{1}{\sqrt{\xi}}\mathrm{erfi}^{-1}(\frac{2p\sqrt{\xi}Z_i(\mathbf{u})}{\sqrt{\pi}e^{-\eta^2/(4\xi)}}) - \frac{\eta}{2\xi}, \tag{6}$$

Where $p$ is a probability sampled from $\mathcal{U}(0,1)$ and $\mathrm{erfi}^{-1}$ is the inverse imaginary error function. Since the sampling requires advanced mathematical skills regarding complex numbers, this sampling method and the extension in general were not explored further.

## 5 Results

### 5.1 Results reproducing original paper

These results are reproduced and visualized, and support the claims made in the original paper regarding identifiability in 2D, iFlow performance, and consistency across latent dimensions.

### 5.1.1 Identifiability in two dimensional space

Some of the 2D visualizations of iFlow compared to iVAE can be seen in figure 1, with more examples in Appendix A.1. Our MCC scores for iFlow are comparable to the original paper, but our results for iVAE are higher, showing comparable or even higher scores than iFlow. These MCC scores do not support the claim that iFlow yields identifiability improvements over iVAE. When looking at the visualizations, it can be argued that iFlow preserves the geometry of the latent space better, since there were cases where iVAE latents retain the manifold of the observations A.3f A.3e. However, contrary to the original paper we found no cases where the latent space of iVAE collapsed. Over 10 seeds,

iFlow gave a mean MCC of 0.77 with standard deviation 0.08, whereas iVAE had a mean MCC of 0.82 with standard deviation 0.06 as shown in Table 2.

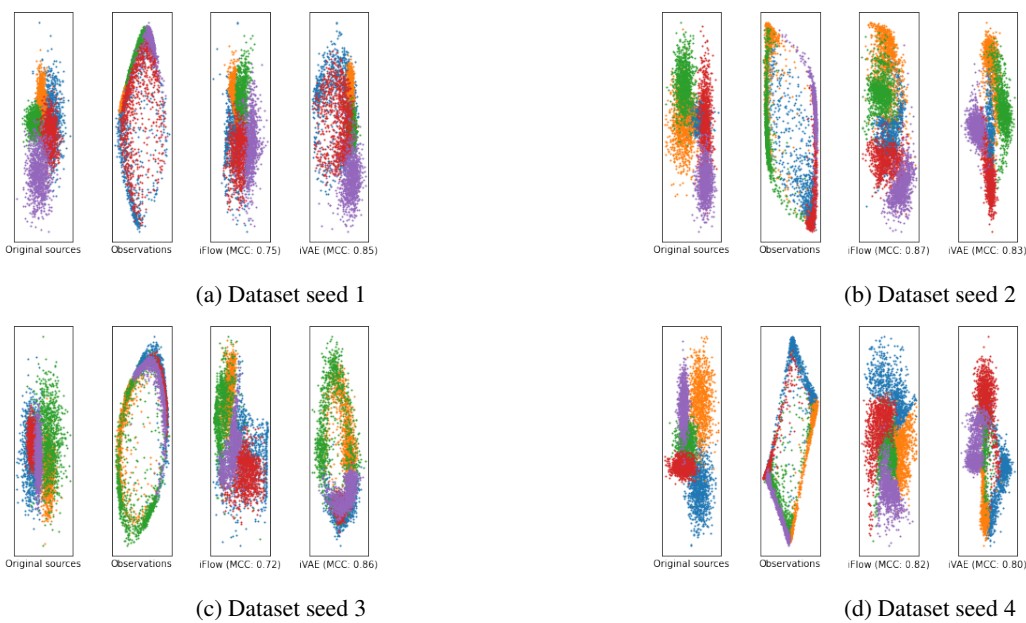

(a) Dataset seed 1           (b) Dataset seed 2

(c) Dataset seed 3           (d) Dataset seed 4

Figure 1: Visualization of 2D cases using different seeds for dataset generation.

### 5.1.2 iFlow performance

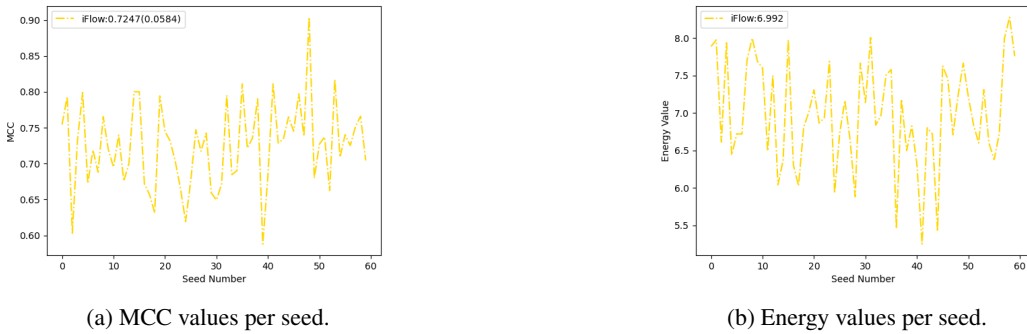

(a) MCC values per seed.          (b) Energy values per seed.

Figure 2: Comparison of identifying performance (MCC) and the energy value (likelihood in logarithm) versus seed number, respectively.

In figure 2 the MCCs and energy values of the iFlow model can be seen across 60 different seeds. Our results are comparable to the results presented in the paper. The means across all seeds are nearly identical as the ones in the paper. This confirms the reproducibility of these results. The MCCs support the first claim of the paper and the energy values support the second claim.

### 5.1.3 Consistency across latent dimensions

Figure 3 is a visualization of the true source latents, plotted against the estimated latents as signals for seed 49. Our measured correlation coefficients are comparable with the original results, however, the signals do not overlap in our reproduction. On this seed, iFlow performs consistently well across all dimensions, while iVAE underperforms in the 3rd and 5th dimension. While this is in favour of the claim that iFlow outperforms iVAE consistently across all dimensions, it has, however, not been studied extensively.

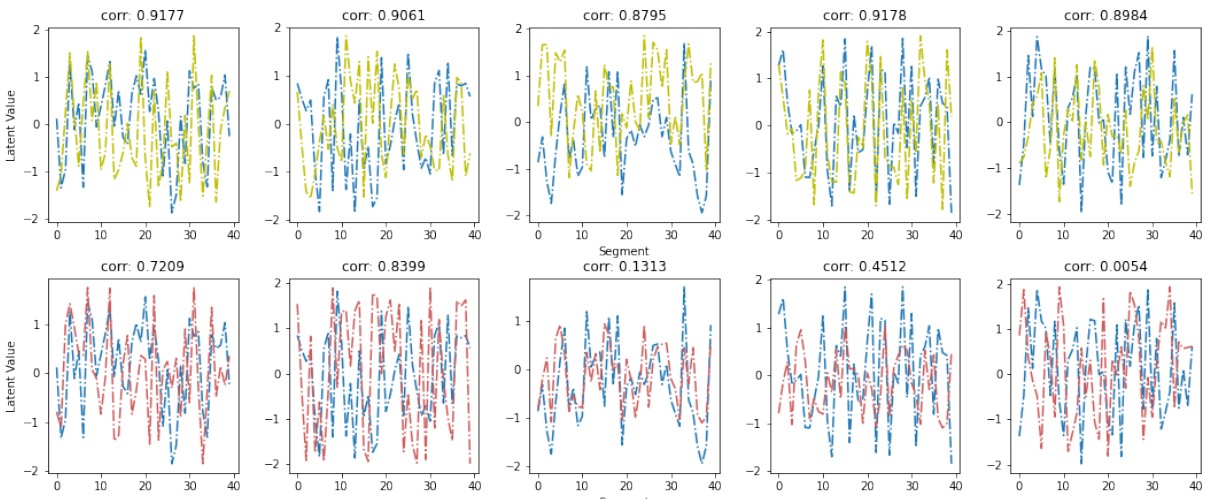

Figure 3: Identifying performance (correlation coefficient) of each dimension of the latent space, using seed 49. The y-axis indicates the mean value of the latent segment, and the x-axis indicates the number of segments.

In order to further study the validity of this claim, we have computed the mean and standard deviation of correlation coefficients in each dimension, for iFlow and iVAE. These results are shown in Table 1. On average, iFlow outperforms iVAE on each dimension by around 0.25 points. Additionally, the standard deviation of the iFlow correlation in each dimension is marginally smaller than iVAE, which shows that iFlow has improved consistency compared to iVAE. Thus, these results show significant support of the third claim.

| model | dim_1 | dim_2 | dim_3 | dim_4 | dim_5 |
|---|---|---|---|---|---|
| iFlow | 0.725($\pm$0.126) | 0.7469($\pm$0.1121) | 0.7076($\pm$0.1225) | 0.7134($\pm$0.1118) | 0.7304($\pm$0.1306) |
| iVAE | 0.4996($\pm$0.257) | 0.4824($\pm$0.259) | 0.4569($\pm$0.2826) | 0.4421($\pm$0.2561) | 0.5002($\pm$0.249) |

Table 1: Average correlation coefficient across each dimension and their standard deviation, measured over 60 seeds.

## 5.2 Results beyond original paper

### 5.2.1 Performance with increased dataset complexity

Table 2 reports the performances of iFlow and iVAE on observations created by increasingly deep mixing MLPs. At 3 layers for the MLP, iVAE outperforms iFlow by 0.045. As the dataset complexity increases, both models lose performance. However, the performance of iFlow declines only marginally with respect to iVAE. At 5 MLP layers, iFlow outperforms iVAE significantly by 0.093 points. This trend supports our claim that iFlow performs more consistently across increased dataset complexities. Additionally, one example visualization is provided in figure 4 which shows the learnt latent space for observations mixed with a 4-layered and 5-layered MLP. On the same seeded dataset, iFlow is able to estimate accurate latents for both mixing complexities, while iVAE fails to estimate latents for the 5-layered MLP, resulting in a collapsed latent space. More visualizations for the 4-layered and 5-layered MLPs are provided in Appendix A.2 and A.3.

| model | 3 layers | 4 layers | 5 layers |
|---|---|---|---|
| iFlow | 0.7727 ($\pm$0.0770) | 0.7374 ($\pm$0.1083) | 0.7332 ($\pm$0.1154) |
| iVAE | 0.8176 ($\pm$0.0561) | 0.6874 ($\pm$0.1188) | 0.6403 ($\pm$0.1458) |

Table 2: Mean and standard deviation of MCC scores measured on 10 seeds.

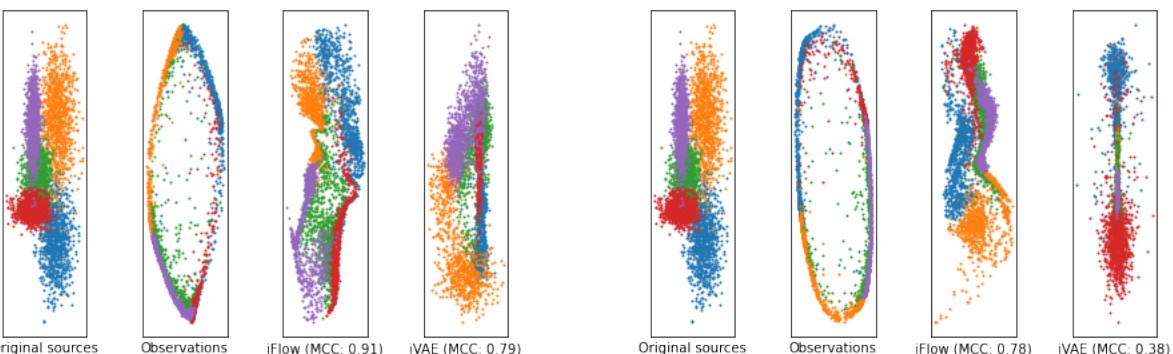

(a) Performance on data mixed by a 4-layer MLP.  (b) Performance on data mixed by a 5-layer MLP.

Figure 4: Visualization of 2D cases with increased data complexity, with seed set to 4.

# 6 Discussion

Firstly, in 2D the MCC scores do not support the claim that iFlow yields identifiability improvements over iVAE. According to the visualizations it can be argued that iFlow preserves the geometry of the latent space better, however there was no case where iVAE collapsed, which is in contradiction to the papers results. One possible explanation is the bug regarding the batch size, which could mean the paper used a batch size of eight rather than the reported 64. Secondly, the results of our iFlow performance study are similar to that of the original paper. This suggests that iFlow indeed yields improved identifiability and improved modelling of data distributions compared to iVAE. Finally, the results of the original paper regarding the last claim were difficult to reproduce. When using the best performing seed, the results suggest in favour of the improved identifiability across dimensions of latent space. But, overall, more research is required to confirm this. Our extended experiments show more stability of iFlow when increasing the data complexity compared to iVAE, which support the claim for iFlow's improved stability across data complexities.

## 6.1 What was easy

Training the models itself was generally straightforward. The code was provided on a GitHub, as well as a script to train models for figure 2. Additionally, descriptions were included in the code for each of the hyperparameters and from the paper it was generally clear which hyperparameter values we had to use. Another convenience was the well-separated and clear file structure of the code, generated data, and logging.

## 6.2 What was difficult

One challenge was that the code contained several blocks of commented lines, where it was initially unclear whether these were deprecated or not. Some of these had to be uncommented and extended with if-else statements in order to make training on both two dimensional and five dimensional data possible. Additionally, there was no code provided for recreating the figures in the paper. The main challenge in recreating these figures ourselves was extracting the correct values for each figure, which required a thorough understanding of both the paper and the code. In attempting to extend the paper, we ran into some additional issues, specifically with the sampling of new data. This is because we were unable to find either a mathematical procedure or implementation to sample from the latent distribution. Creating such a sampling procedure requires advanced mathematical knowledge. As a result, implementing iFlow as a generative model is not trivial.

## 6.3 Communication with original authors

We have made attempts to get in contact with the original authors through their e-mail, but have been unsuccessful in doing so.

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

# Appendices

## A  2D visualization

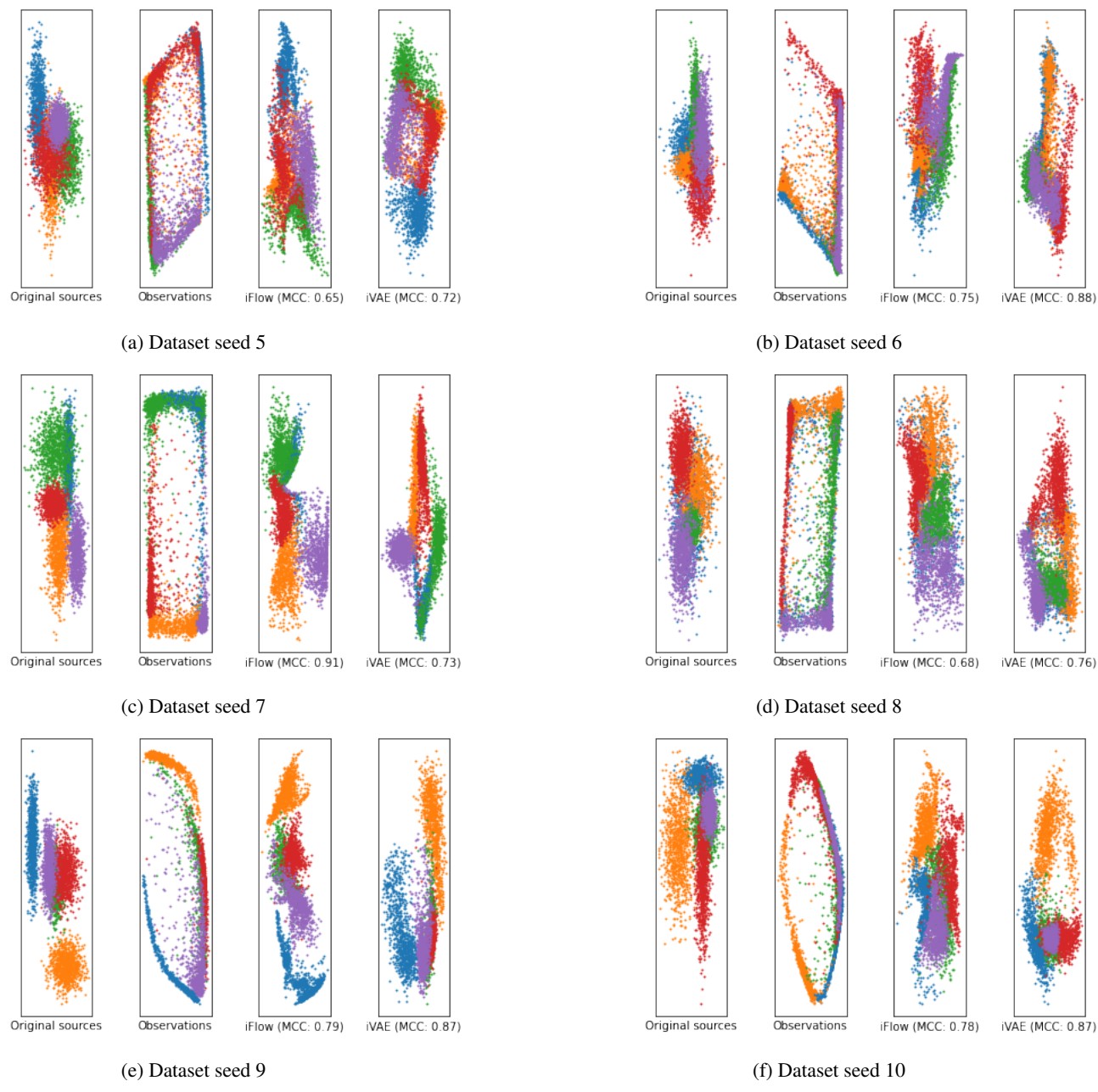

(a) Dataset seed 5

(b) Dataset seed 6

(c) Dataset seed 7

(d) Dataset seed 8

(e) Dataset seed 9

(f) Dataset seed 10

Figure A.1: Visualization of 2D cases using different seeds for dataset generation and 3 mixing layers, like in the original paper.

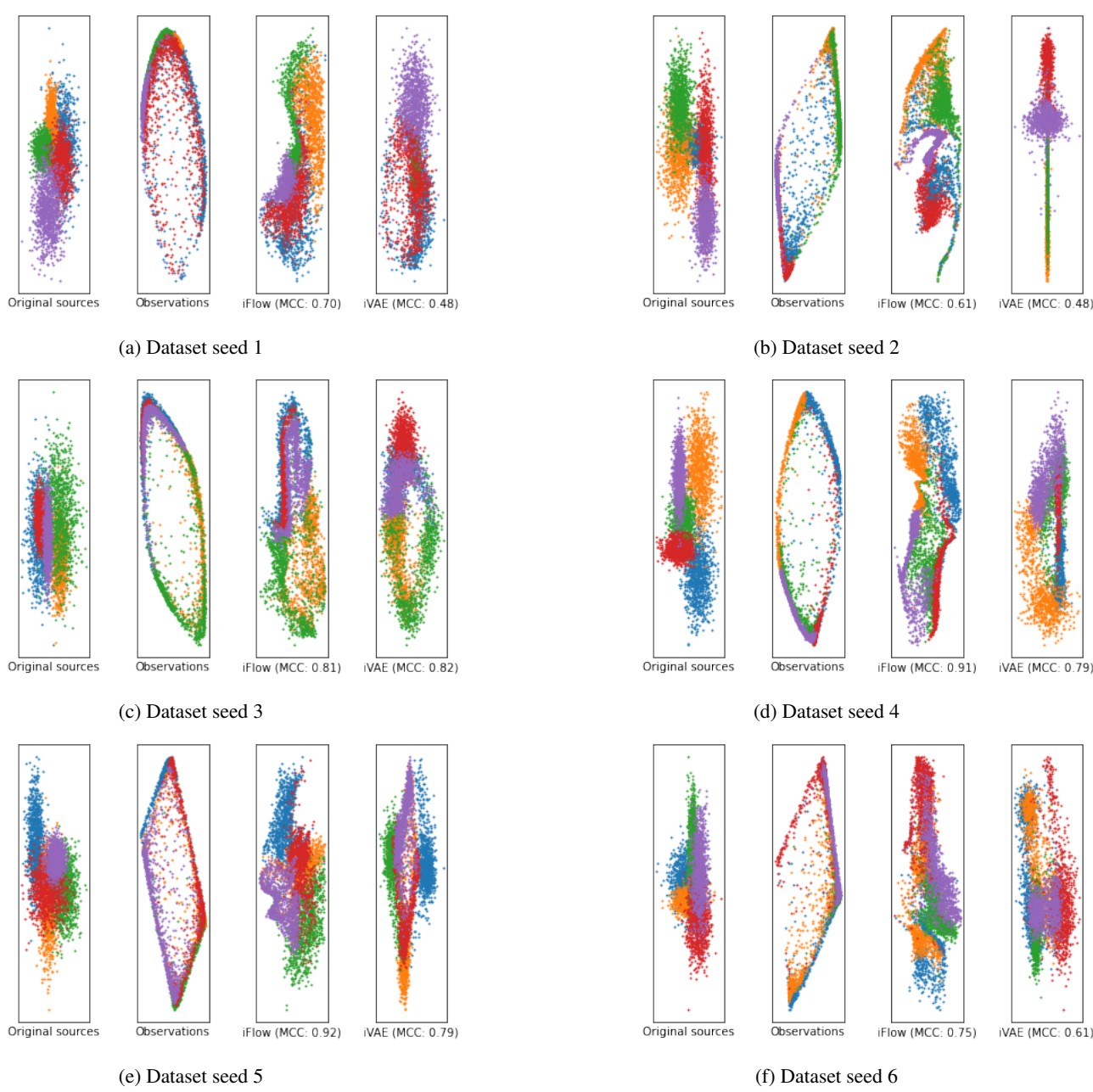

(a) Dataset seed 1

(b) Dataset seed 2

(c) Dataset seed 3

(d) Dataset seed 4

(e) Dataset seed 5

(f) Dataset seed 6

Figure A.2: Visualization of 2D cases using different seeds for dataset generation and 4 mixing layers.

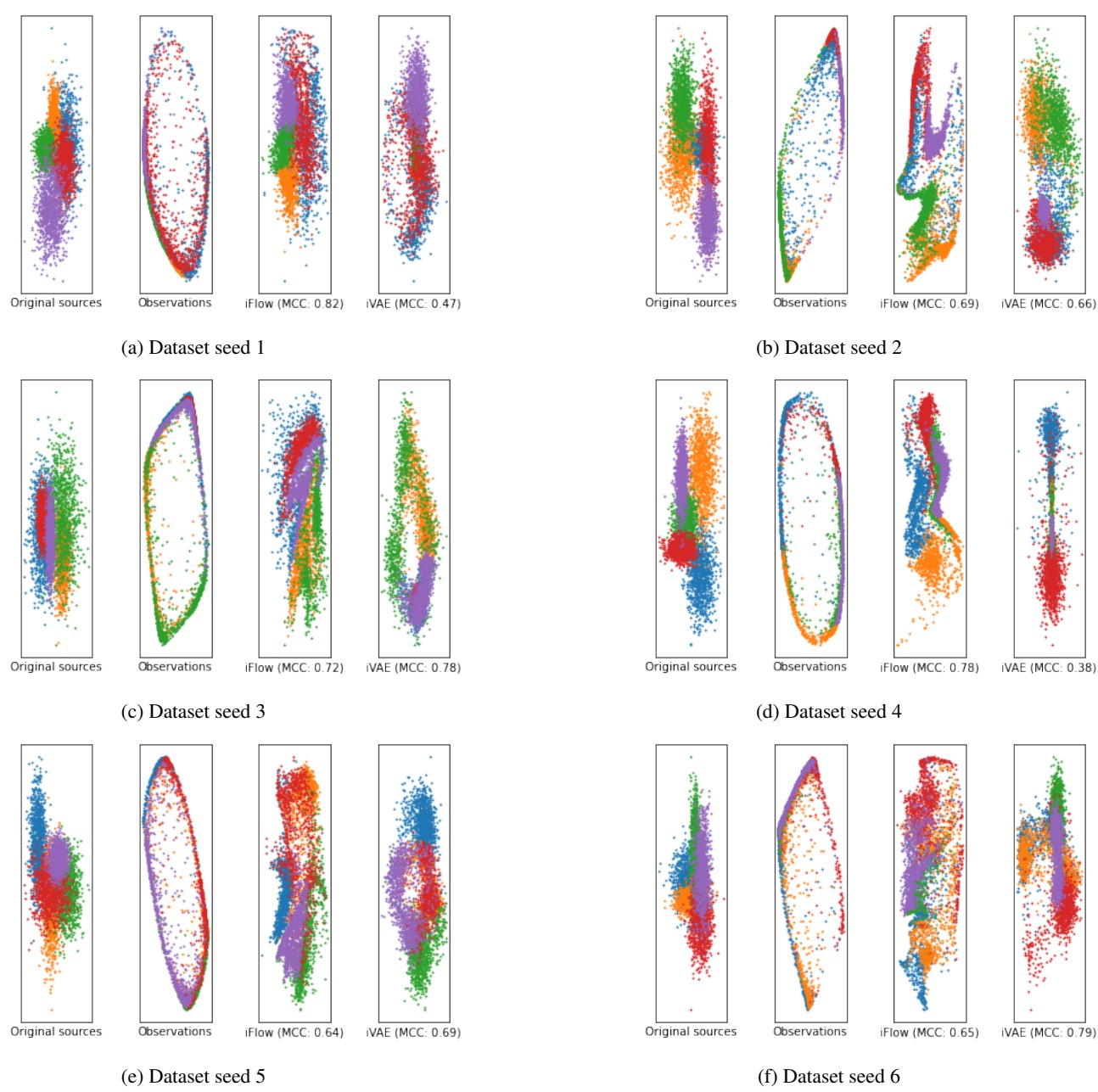

Figure A.3: Visualization of 2D cases using different seeds for dataset generation and 5 mixing layers.

