# OpenReview forum: "[Re] Identifying Through Flows for Recovering Latent Representations"
_ML_Reproducibility_Challenge/2020 — Reject_

### Official Review · AnonReviewer2 · 2021-03-02

**Rating:** 6
**Confidence:** 3

**Review:**

This work aims at reproducing the iFlow method proposed in "Identifying Through Flows for Recovering Latent Representations". The original paper made some claims against the baseline approach iVAE, and this work reran the experiments and validated some of its claims, but not all: in particular, this work showed that in the 2D case, iVAE does not have mode collapsing issues as claimed by the original paper and actually outperforms iFlow in terms of MCC. Other than this, this work is able to reproduce the major claims by the original paper.

Pros:
1. This work conducted many experiments and was able to rerun most of the experiments in the original paper despite some experiments needing changes to the code.

Cons:
1. In section 6 the authors suspected that "One possible explanation is the bug regarding the batch size, which could mean the paper used a batch size of eight rather than the reported 64." Is there evidence that under batch size 8 VAE gets worse performance and occasionally gets mode collapsing?

2. I like the extension to the MNIST case, but unfortunately, this work didn't propose a feasible implementation.

3. In terms of presentation style, the captions of figure 1 are too small, and it's not very clear what are the two rows of figure 3. Is the top row iVAE and the bottom iFlow?

Overall, this work has reproduced the major claims of the original paper, and also showed some potential bugs of the original paper in the 2D case. I am leaning towards its acceptance.

**Familiar With The Original Paper:**

I have not read the original paper

**Reproducibility Summary:**

Report has summary

---

### Official Review · AnonReviewer3 · 2021-03-02
**Overall good report with an extended evaluation of the original paper**

**Rating:** 7
**Confidence:** 4

**Review:**

Reproducibility Summary

The authors did an excellent job of explaining the summary of the report. They also mention a detailed description of the reproduced results and the difficulty faced during the reproducibility of the original paper.

Scope of reproducibility

The authors have clearly stated the scope of reproducibility with clarity in the claims they learned from the original paper with further details explaining the clams.

Code

From my understanding, the code was provided by the original authors with detailed information about the hyper-parameter search. But the authors do a commendable job in preparing the code-base for figures and thoroughly checking the code for any possible scope of improvement.

Communication with original authors

As mentioned by the authors they were not able to get clarification from the original authors. It would be great if they could share the doubts they had during the reproducibility which were not clarified by the original authors.

Hyperparameter Search

The original authors have described the hyper-parameters needed for the successful replication of the numbers. But the author’s effort of verifying every seed and python version to get the closest results is really commendable and also making the code publicly available is a crucial step for advancing the current work.

Ablation Study

The authors did a great job of showing ablation for both iFlow and iVae methods.

Discussion on results

There is a discussion section mentioned in the paper that repeats how the experiments support the claims mentioned in the original paper. It would have been great to see if the authors can provide a thorough description of the method strengths and weakness and why the experiments can support the claim.

Results beyond the paper

The authors have mentioned a section in which they mention the experiment with a real dataset to check if the claim still holds. It is also nice to see the difficultly faced while experimenting with a new dataset. It would have been great to see if both the paragraphs (sec. 4.4.7 sec. 6.2) would be structured more appropriately for easier readability.

Overall organization and clarity

I found the report to be well organized and extremely easy to read. The plots, however, could be slightly updated to improve readability.


**Familiar With The Original Paper:**

I have read the original paper

**Reproducibility Summary:**

Report has summary

---

### Decision · Program_Chairs · 2021-03-31

**Decision:**

Reject

**Comment:**

The report is non-anonymized.
Otherwise, while the results are interesting, the report isn't well-written enough to explain the results in an adequate manner.